# Exhaled Breath Analysis in Lymphangioleiomyomatosis by Real-Time Proton Mass Spectrometry

**DOI:** 10.3390/ijms26136005

**Published:** 2025-06-23

**Authors:** Malika Mustafina, Artemiy Silantyev, Marina Makarova, Aleksandr Suvorov, Alexander Chernyak, Zhanna Naumenko, Pavel Pakhomov, Ekaterina Pershina, Olga Suvorova, Anna Shmidt, Anastasia Gordeeva, Maria Vergun, Olesya Bahankova, Daria Gognieva, Aleksandra Bykova, Andrey Belevskiy, Sergey Avdeev, Vladimir Betelin, Philipp Kopylov

**Affiliations:** 1Department of Cardiology, Functional and Ultrasound Diagnostics, I.M. Sechenov First Moscow State Medical University (Sechenov University), 119991 Moscow, Russiagognieva_d_g@staff.sechenov.ru (D.G.); kopylov_f_yu@staff.sechenov.ru (P.K.); 2Pulmonology Research Institute under the Federal Medical and Biological Agency of Russia, 115682 Moscow, Russia; mma123@list.ru (M.M.); achi2000@mail.ru (A.C.); naumenko_janna@mail.ru (Z.N.);; 3Research Institute for Systemic Analysis of the Russian Academy of Sciences, 117218 Moscow, Russia; 4World-Class Research Center “Digital Biodesign and Personalized Healthcare”, I.M. Sechenov First Moscow State Medical University (Sechenov University), 119991 Moscow, Russia; artsilan@gmail.com (A.S.); suvorov_a_yu_1@staff.sechenov.ru (A.S.);; 5N.I. Pirogov Russian National Research Medical University, 1 Ostrovityanova str., Bldg. 6, 117513 Moscow, Russia; 6National Medical Research Center of Otorhinolaryngology, Federal Medical and Biological Agency of Russia, 123182 Moscow, Russia; pavel.v.pakhomov@gmail.com; 7First Moscow City Hospital Named After N.I. Pirogov, 119049 Moscow, Russia; 8Pulmonology Department, I.M. Sechenov First Moscow State Medical University (Sechenov University), 119991 Moscow, Russia; olga.a.suvorova@mail.ru (O.S.); a_e_schmidt@mail.ru (A.S.); gordeeva.anast.aleks@gmail.com (A.G.)

**Keywords:** lymphangioleiomyomatosis, breath analysis, volatile organic compounds, proton mass spectrometry

## Abstract

Lymphangioleiomyomatosis (LAM) is a rare progressive disease that affects women of reproductive age and is characterized by cystic lung destruction, airflow obstruction, and lymphatic dysfunction. Current diagnostic methods are costly or lack sufficient specificity, highlighting the need for novel non-invasive approaches. Exhaled breath analysis using real-time proton mass spectrometry (PTR-MS) presents a promising strategy for identifying disease-specific volatile organic compounds (VOCs). This cross-sectional study analyzed exhaled breath samples from 51 LAM patients and 51 age- and sex-matched healthy controls. PTR-time-of-flight mass spectrometry (PTR-TOF-MS) was employed to identify VOC signatures associated with LAM. Data preprocessing, feature selection, and statistical analyses were performed using machine learning models, including gradient boosting classifiers (XGBoost), to identify predictive biomarkers of LAM and its complications. We identified several VOCs as potential biomarkers of LAM, including *m*/*z* = 90.06 (lactic acid) and *m*/*z* = 113.13. VOCs predictive of disease complications included *m*/*z* = 49.00 (methanethiol), *m*/*z* = 48.04 (O-methylhydroxylamine), and *m*/*z* = 129.07, which correlated with pneumothorax, obstructive ventilation disorders, and radiological findings of lung cysts and bronchial narrowing. The classifier incorporating these biomarkers demonstrated high diagnostic accuracy (AUC = 0.922). This study provides the first evidence that exhaled breath VOC profiling can serve as a non-invasive additional tool for diagnosing LAM and predicting its complications. These findings warrant further validation in larger cohorts to refine biomarker specificity and explore their clinical applications in disease monitoring and personalized treatment strategies.

## 1. Introduction

Lymphangioleiomyomatosis (LAM) is a rare, progressive, multisystem disorder that predominantly affects women of reproductive age. It is characterized by the abnormal proliferation of smooth muscle-like cells (LAM cells), leading to cystic lung destruction, airflow obstruction, and lymphatic dysfunction. The pathogenesis of LAM is closely linked to mutations in the tuberous sclerosis complex (*TSC*) genes, *TSC1* and *TSC2* genes, which encode hamartin and tuberin, respectively, critical components of the mammalian target of rapamycin pathway (mTOR) signaling pathway [1]. This dysregulation results in enhanced cellular proliferation, survival, and migration, which are hallmarks of the disease. Despite these molecular insights, the clinical management of LAM remains challenging due to variability in disease progression and the limited specificity of existing diagnostic biomarkers [2].

Early detection and monitoring are critical for managing LAM, particularly as therapies such as mTOR inhibitors (e.g., sirolimus) have shown efficacy in stabilizing lung function [3]. However, current diagnostic tools, such as high-resolution computed tomography (HRCT) imaging and vascular endothelial growth factor–D (VEGF-D) serum levels, are either costly or insufficiently specific for early-stage disease [4]. This underscores the need for novel, non-invasive diagnostic approaches that can provide real-time, individualized insights into the metabolic and pathological processes of the disease.

One promising avenue in this regard is exhaled breath analysis, a non-invasive technique capable of reflecting underlying biochemical alterations through the profiling of volatile organic compounds (VOCs). Human breath contains a rich mixture of VOCs that are products of normal and pathological metabolic processes. These molecules, detectable in parts per billion or trillion, have shown utility as biomarkers in various respiratory conditions, including asthma, chronic obstructive pulmonary disease (COPD), and lung cancer [5].

Among analytical methods, proton transfer reaction mass spectrometry (PTR-MS) has emerged as a leading tool for VOC analysis. PTR-MS enables real-time, high-sensitivity quantification of VOCs without the need for extensive sample preparation, making it particularly suited for clinical applications [6]. Studies employing PTR-MS have demonstrated its ability to discriminate between healthy individuals and patients with various lung diseases by identifying specific VOC signatures associated with oxidative stress, inflammation, and tissue remodeling [7,8,9].

In the context of LAM, the reliance of the disease on dysregulated mTOR signaling and increased oxidative stress may generate unique VOC profiles detectable in exhaled breath. By leveraging PTR-MS, it is possible to uncover these metabolic fingerprints, thereby providing valuable insights into the pathophysiological processes driving LAM progression. Moreover, such breath-based diagnostics could complement existing biomarkers like VEGF-D, offering a more comprehensive and less invasive approach to disease monitoring.

This study investigates the application of PTR-MS to the analysis of exhaled breath in patients with LAM. Specifically, we aim to characterize the unique VOC profiles associated with LAM and advance the understanding of the possible metabolic and biochemical disruptions underlying LAM pathophysiology.

## 2. Results

### 2.1. Baseline Characteristics

Patients with LAM and healthy control women aged 27 to 75 years were included in this cross-sectional study. Female patients and volunteers from the control group were matched in age, anthropometric data, and smoking status. Obstructive ventilation disorders were identified in 26 (51.0%) patients, and a decrease in diffusing capacity of the lungs for carbon monoxide (DLCO) in 35 (68.6%), in contrast to the control group (*p* < 0.001). The main characteristics of all participants are presented in Table 1.

### 2.2. Identification of VOCs as Predictors of LAM

Fifty-one patients with LAM and 51 healthy control women performed a quiet breathing maneuver and a three-time forced expiratory maneuver, resulting in 408 exhaled air samples. A total of 198 features in all samples were determined using PTR-TOF-MS in patients with LAM and healthy volunteers. Seventy-five of them turned out to be significant clinical and functional predictors.

Selection using the XGBoost algorithm identified the following predictors of LAM: *m*/*z* = 90.06 (lactic acid) and not indicated *m*/*z* = 50.00, *m*/*z* = 82.07, *m*/*z* = 99.07, *m*/*z* = 113.13, *m*/*z* = 127.00, *m*/*z* = 150.010, and *m*/*z* = 329.82, both in quiet breathing and forced expiratory samples, with AUC = 0.922 (sensitivity—0.844; specificity—0.849). Feature importances are presented in Table 2.

### 2.3. Identification of VOCs as Predictors of Respiratory Disorders, Radiological Changes, and Significant Clinical Outcomes

Almost half of the patients with LAM had a history of pneumothorax (25 patients, 49%), which was defined as a clinical outcome. The most significant predictors of this clinical outcome from those VOCs that were able to be annotated were *m*/*z* = 103.08 (isopropyl acetate) and *m*/*z* = 133.10 (2-ethoxyethyl acetate); feature importances are presented in Table 3. The results of the composite model for clinical and functional outcomes based on selected VOCs during normal and forced exhalation are presented in Appendix A.

Obstructive disorders were defined as a decrease in the ratio forced expiratory volume in 1 s to forced vital capacity (FEV_1_/FVC) and FEV_1_ below the lower limit of normal, and air trapping as an increase in residual volume (RV) and in the ratio RV to total lung capacity (RV/TLC) above the upper limit of normal. In addition, decreased pulmonary gas exchange function was defined as a decrease in DLCO below the lower limit of normal. For these functional outcomes, significant predictors were identified using the XGBoost algorithm (Table 3).

Using chest HRCT, cysts in the lungs were identified in all patients with LAM. With a comprehensive analysis of tomograms, the mean volume of cysts in both lungs was 22.3 ± 16.5% of the total volume of the lungs. In addition, the average lumen area of the distal part of the B1 and B10 bronchi of the right lung in patients with LAM was analyzed, and it turned out to be reduced when compared with those described in the literature in healthy subjects [10]. The analysis also identified VOCs as predictors of these radiological outcomes (Table 3).

Performance metrics for normal and forced breathing for different clinical and functional endpoints are presented in Appendix A. Summarized importance of features for LAM diagnosis, for pneumothorax, for respiratory outcomes and for radiographic changes in LAM are presented in Appendix A, respectively.

Direct comparison of differences for the most significant biomarkers identified in our study shows significant differences and is presented in Table 4. 

Thus, the most important VOCs as predictors are *m*/*z* = 49.00 (methanethiol), *m*/*z* = 48.04 (O-methylhydroxylamine), and *m*/*z* = 129.07. Their feature importance increased during both normal quiet breathing and forced expiratory maneuvers across several clinical and functional outcomes.

## 3. Discussion

This is the first study to analyze the composition of exhaled air in patients with LAM. Despite the fact that the sample is relatively small for statistical analysis, 51 patients took part in the study, which is quite a lot for an orphan disease.

For the first time, specific biomarkers of LAM have been identified using PTR-TOF-MS: *m*/*z* = 90.06 (possible, lactic acid), and *m*/*z* = 113.13, both during quiet breathing and during the forced expiratory maneuver. Lactic acid may be a consequence of anaerobic glycolysis in hypoxic areas of the lung. In addition, this metabolite may promote angiogenesis in LAM. Elevated lactate in breath may correlate with disease progression, similar to its role in other respiratory diseases (e.g., COPD) where lactate levels reflect exacerbation severity [11]. When analyzing predictors of not only the diagnosis of LAM, but also clinical and functional complications of the disease (number of pneumothoraxes; respiratory dysfunction in the form of obstructive disorders, air traps, and gas exchange disorders; or the presence of cysts during X-ray examination), the most significant VOCs as predictors were *m*/*z* = 49.00 (methanethiol), *m*/*z* = 48.04 (O-methylhydroxylamine), and *m*/*z* = 129.07.

According to the described pathogenesis of LAM [12], it can be assumed that the identified VOCs are products of one or another part of the pathological process of the disease, as shown in Figure 1. Given that LAM is driven by *TSC1/2* mutations and consequent hyperactivation of the mTORC1 pathway, the observed VOC profile may reflect downstream perturbations in lipid peroxidation, amino acid metabolism, and oxidative stress responses.

One of these links in pathogenesis is oxidative stress and lipid peroxidation, which accompany all chronic inflammatory processes. In LAM, dysregulated lipid metabolism has been observed, which may exacerbate oxidative stress. The accumulation of lipids within LAM cells can render them more susceptible to peroxidation. These lipid peroxidation products can form adducts with proteins, nucleic acids, and other cellular components, disrupting their normal function and contributing to disease progression. Of the VOCs we identified as significant predictors of LAM and its complications, probable products of oxidative stress and inflammation include O-methylhydroxylamine, octene, isomers of oxononanoic acid, and methyl ethyl ketone. Lipid peroxidation in LAM can lead to the formation of reactive aldehydes, examples of which are those found in our study, including methyl ethyl ketone and ethers (2-ethoxyethyl acetate and isopropyl acetate). Methyl ethyl ketone turned out to be a significant predictor of a reduced lumen area of the distal bronchi, that is, it is probably associated with obstructive ventilation disorders characteristic of LAM. According to the literature, this metabolite has been described as significant for *Pseudomonas aeruginosa* in cystic fibrosis [9,13].

One of the key molecular mechanisms underlying LAM pathogenesis is the dysregulation of proteolytic enzymes, particularly matrix metalloproteinases (MMPs) and serine proteases, which contribute to extracellular matrix (ECM) degradation and facilitate the invasive behavior of LAM cells. The activation of pro-MMPs occurs via proteolytic cleavage by other proteases (e.g., plasmin or cathepsins) or oxidative stress-induced conformational changes. Degradation of elastin by MMP-2, MMP-9, and cathepsins weakens alveolar structures, leading to the formation of cystic airspaces characteristic of LAM [14]. Products of the dysregulation of MMPs may be metabolites N-methylacetamide and 2-nonenal in exhaled air, which turned out to be significant predictors of obstructive ventilation disorders and air traps in patients with LAM. In a study by Lee et al., an N-methylacetamide derivative was found to have anti-inflammatory and protective effects against acute lung tissue injury [15].

Activation of proteases and MMPs also leads to the catabolism of sulfur-containing amino acids. Possible products of altered sulfur metabolism are those we discovered: 2-butanethiol, dimethyl sulfide, and methanethiol in exhaled air. These VOCs were found to be predictive of obstruction (FEV_1_ < LLN) and LAM-specific radiographic manifestations (lung cysts and small airway obstruction). Studies have previously reported dysregulation of sulfur metabolism as detected by sulfur-containing VOCs in patients with cancer compounds [16]. These metabolites include methanethiol, the level of which was increased in various cancers [16]. It is likely that pathological processes in LAM are similar to oncology. In particular, a study by Klarquist et al. revealed the susceptibility of in vitro cultured LAM cells to reactive melanoma cytotoxins from T cells [17].

We can hypothesize a possible relationship between mTOR-induced cellular changes and specific VOCs in exhaled air in patients with LAM. mTOR hyperactivation in LAM cells promotes metabolic reprogramming toward lipogenesis and defective autophagy, leading to the accumulation of lipid-derived oxidation products (e.g., 2-nonenal) and incomplete metabolite clearance. Furthermore, mTOR’s suppression of ketolysis may explain elevated methyl ethyl ketone levels. Further studies should explore whether these VOCs correlate with mTOR activity in LAM cell models and whether their suppression via mTOR inhibitors (e.g., sirolimus) normalizes exhaled breath profiles. This could solidify their role as non-invasive biomarkers of disease progression or therapeutic response.

Despite the promising findings of this study, several limitations must be acknowledged. First, the sample size may have been insufficient to fully capture the variability of exhaled VOCs in LAM and control populations. A larger cohort would enhance the statistical power and generalizability of the results. Additionally, the cross-sectional study design precludes conclusions about disease progression or the temporal stability of the identified biomarkers. It is likely that future longitudinal studies will assess whether these VOC characteristics change over time or in response to treatment. Unfortunately, we were unable to identify an important number of VOCs (apparently protein-lipid fragments) using the IONICON library, the Human Metabolome Database (HMDB), and literature data. This could lead to incomplete and partial errors in the annotation of VOCs. Perhaps, multi-modal approaches (gas chromatography and tandem mass spectrometry) and standardized databases could improve reliability. In addition, despite the observance of certain conditions for collecting material (described above) for the analysis of exhaled air, we did not take into account the different medications taken by patients with LAM. The study cohort comprised a substantial proportion of treatment-naïve patients, defined as individuals assessed immediately post-diagnosis and prior to the initiation of prescribed therapeutic interventions. In this regard, only 20 patients took specific therapy with mTOR inhibitors, and due to the small sample size, it is not possible to statistically assess the contribution of therapy to the composition of exhaled air in this study. Future longitudinal or multi-center studies with larger treated cohorts are needed to evaluate the impact of mTOR inhibitors on exhaled breath profiles. Finally, in our study, we did not evaluate the relationship of the important LAM biomarker VEGF with detected VOCs in exhaled air, which may be the subject of our future studies. Additionally, future work should integrate serum VEGF measurements with breathomics to uncover potential mechanistic links between angiogenic signaling and metabolic VOC signatures in LAM.

## 4. Materials and Methods

### 4.1. Study Design and Participants

Patients with LAM and controls matched one-to-one for sex and age were included in this observational cross-sectional study. Patients were recruited from the Register of Patients with Lymphangioleiomyomatosis of the Russian Respiratory Society, Moscow. The diagnosis of LAM was established by the presence of respiratory disorders (progressive shortness of breath, recurrent pneumothorax), accompanied by specific changes on HRCT, as well as by extrapulmonary manifestations of the disease (renal angiomyolipomas, abdominal lymphadenopathy and lymphangioleiomyomas, and chylous effusion) [18]. The criteria for exclusion from the study were age under 18 years, pregnancy, and refusal to sign an informed consent. Healthy female volunteers were recruited from traumatology department patients of Sechenov University, Moscow, with the following exclusion criteria: the presence of acute or chronic bronchopulmonary diseases and pregnancy.

Respiratory function assessment (forced spirometry, body plethysmography, and measurement of carbon monoxide uptake in the lungs using the single-breath technique) was performed according to the standards of the American Thoracic Society and the European Respiratory Society [19,20,21,22]. The Global Lung Function Initiative set of reference values was used as predicted [22]. The results are expressed as a percentage of the predicted values.

Chest HRCT was performed on a 640-medium using the Canon Aquilion One CT scanner Genesis (Canon Medical Systems Corporation, Otawara-shi, Japan) with 64 slices. Images were reconstructed using a sharp reconstruction kernel for parenchyma and viewed at window settings optimized for the assessment of the lung parenchyma (window width: 1500 HU; window level: −500 HU). The Thoracic Volume Computer-Assisted Reading (VCAR) (Advantage Workstation version 4.5, GE HealthCare, Chicago, IL, USA) was used for efficient, accurate analysis and visualization of lung and airway imaging using HRCT scans. The HRCT scans of patients with LAM by pre-setting a threshold value of Hounsfield Units were used to obtain a segmentation of both lungs and a quantitative evaluation of cyst (−1024/−950; blue) and healthy lung parenchyma (−950/−703; yellow). The airway diameter and length for B1 and B10 bronchi in the right lung were identified using Thoracic VCAR.

This study was approved by the I.M. Sechenov First Moscow State Medical University ethical committee (Protocol No. 02-23 of 26 January 2023). This study was performed according to the Declaration of Helsinki and registered at ClinicalTrials.gov (NCT05727852). Written informed consent was obtained from all study participants.

### 4.2. Collection of Exhaled Breath and Measurement of VOCs

Exhaled breath was analyzed using PTR-TOF-MS (Ionicon, Innsbruck, Austria). The method of collecting material is described in detail in our previous works [9,23]. Analysis of exhaled air in patients with LAM and healthy volunteers was carried out on an empty stomach, after brushing teeth with a toothbrush, at the same time of day, and in the first half of the day to exclude the influence of circadian rhythms and food and drink intake. Patients with LAM and the control group exhaled into a disposable sterile mouthpiece for 1 min—quiet breathing (12 to 16 exhalation cycles were analyzed), without holding the breath. In addition, the subjects performed forced exhalation three times for 20–40 s without holding their breath, and the highest quality sample was collected.

Exhaled air was analyzed in real time using Ultra-Fast PTR-TOF 1000 (Ionicon, Innsbruck, Austria) mass spectrometry with Buffered End-Tidal Breath Sampling Inlet option (Ionicon, Innsbruck, Austria). Full scan mass spectra were obtained in 10–685 *m*/*z* with a scan time of 1000 ms and primary ion H_3_O^+^. The temperature of T-Drift and T-Inlet was 80 °C. The patient breathed into the sampler through a disposable mouthpiece for 1 min (during this time, depending on the respiratory rate, from 12 to 16 exhalation cycles are analyzed).

### 4.3. Data Processing

Raw data preprocessing was conducted using a custom-developed algorithm written in Python (3.9), incorporating the h5py (3.9.0), scipy (1.11.1), and pandas (2.0.3) libraries. Each sample underwent recalibration for every spectrum, utilizing three specific ions from the Ionicon Permeation Source for Calibration (PerMaSCal) system: 21.0220, 203.94299, and 330.85 *m*/*z*. To ensure data quality, capnostat readings were analyzed. If the capnostat signal of the sample exceeded 3.5 units, it was excluded from further analysis. However, capnostat data were not directly used for assessing inhalation and exhalation patterns due to its delayed response compared to the water adduct signal of the hydronium ion.

For every spectrum in a sample, extracted ion currents were identified for three biogenic ions: isoprene ([M + H]^+^, *m*/*z* = 69.07), dimethyl sulfide (*m*/*z* = 63.02), and 1,2-butadiene (*m*/*z* = 55.03). Additionally, capnostat readings and signals from an ion at *m*/*z* = 37.038, representing the water adduct with the hydronium ion ([H_2_O + H_3_O]^+^), were recorded. The response of this ion was indicative of exhaled moisture levels and helped track inhalation–exhalation cycles during breathing. Only spectra in which the water adduct signal surpassed a threshold of 2 × 10^5^ cps and exhibited a local maximum for at least two of the three biogenic ions were included in further processing.

Additional quality control measures were applied to the collected samples. The mass errors of calibrant ions from the PerMaSCal system at 21.022, 203.94, and 330.85 *m*/*z* were verified, with acceptable error margins below 100 ppm, typically ranging between 10 and 30 ppm. Moreover, the number of local maxima corresponding to respiratory cycles was assessed. Samples displaying fewer than three complete inhalation–exhalation cycles were disregarded.

To refine the spectral data, selected spectra within each sample were summed, averaged, and smoothed using the Savitzky–Golay filter. Following this, ion peaks were extracted and filtered. Identified ions were aligned across different samples if they appeared in over 50% of cases. The resulting mass-to-charge ratio (*m*/*z*) dataset facilitated signal extraction within a ±0.015–0.4 *m*/*z* range.

Finally, for each patient, the extracted data were averaged, normalized to the cluster heavy water isotope signal with the hydroxonium ion ([D_2_O + H_3_O]^+^), and prepared for subsequent statistical evaluation.

### 4.4. VOCs Annotation

For features displaying significant differences, identification was carried out based on the observed ion mass-to-charge ratio. The reference database was compiled using two proprietary Ionicon ion libraries—one containing 300 factory-calibrated masses and the other comprising 1000 factory masses—along with the HMDB, with chemical compounds documented in the scientific literature as chronic respiratory disease markers. The maximum allowable deviation for annotation was set at ±200 ppm; it was assumed that the identified VOCs were recorded in the protonated form as [M + H]+.

### 4.5. Statistical Data Analysis

#### 4.5.1. Descriptive Statistics

All statistical methods used were the same as those used in our previous work involving patients with cystic fibrosis [9]. For quantitative variables, basic exploratory data analysis was performed: metrics such as distribution characteristics (assessed via the Shapiro-Wilk test), mean, standard deviation, median, interquartile range, 95% confidence interval, and the observed minimum and maximum values were computed. For categorical and qualitative features, the absolute count and rates were estimated.

Comparative analysis for normally distributed quantitative features was conducted using Welch’s *t*-test (for 2 groups) or ANOVA (for more than 2 groups) followed by pairwise group comparisons; for non-normally distributed quantitative features, the Mann–Whitney U-test (for 2 groups) or the Kruskal–Wallis test (for more than 2 groups) was used. Comparative analysis of categorical and qualitative features was performed using Pearson’s chi-square test, or, when inapplicable, Fisher’s exact test. The significance level was set at 0.05.

#### 4.5.2. Identifying VOC Predictors and Their Relationship to Endpoints

To determine predictive markers, a cross-validation approach was applied, during which data transformation and classifier training were conducted to evaluate predictor significance within a single model. All endpoints were examined separately for both forced and regular exhalation. The analysis involved repeated cross-validation, data normalization, and training XGBoost for each dataset split for binary endpoints or regressors in the case of quantitative endpoints. The effectiveness of the model and the relative importance of features were evaluated using the area under the receiver operating characteristic (ROC) curve (AUC) for classifiers, and the coefficient of determination, maximal error, explained variance, and root mean squared error for regressors were assessed.

Only VOCs were considered as potential predictors, with calibration molecules and substances with a mass-to-charge ratio (*m*/*z*) below 42 being excluded from the analysis.

#### 4.5.3. Feature Selection Process

Given the limited sample size, a resampling technique was implemented, wherein 2/3 of the dataset was randomly selected 1000 times. During each iteration, preprocessing steps, including data normalization and iterative imputation via Bayesian ridge regression, were performed. The feature importance of the XGBoost classifier or regressor was determined in each iteration. The median feature importance score across all 1000 iterations was calculated for each VOC and ranked in descending order. This selection process was carried out independently for forced and regular breathing. For each endpoint, the 30 VOCs with the highest median scores from both respiratory conditions were compared to identify overlapping features. VOCs common to both breathing conditions were retained for further evaluation.

#### 4.5.4. Examining the Association Between Selected VOCs and Outcomes

For each endpoint, the common VOCs were utilized to construct new XGBoost classifiers using the entire dataset, applying leave-one-out cross-validation separately for forced and regular breathing maneuvers. The probability of each outcome’s occurrence was calculated to assess classifier performance, including sensitivity, specificity, and predictive values (both positive and negative) based on a threshold probability of 0.5. Furthermore, in the best-performing models obtained from cross-validation, feature importance was recalculated, considering only the common VOCs. This refinement allowed the identification of VOCs with consistently high feature importance scores and enabled direct comparison of their predictive value across forced and regular breathing scenarios.

## 5. Conclusions

This study successfully identified specific biomarkers of LAM in exhaled breath using real-time proton mass spectrometry. The findings provide valuable insights into the metabolic alterations associated with LAM and highlight the potential of non-invasive breath analysis as a diagnostic tool. Additionally, we identified predictive markers for complications, including pneumothorax, impaired respiratory function, and characteristic radiological findings such as pulmonary cysts and a decrease in the average lumen area of the distal part of the bronchi, as revealed by computed tomography. These biomarkers may serve as valuable indicators for disease monitoring and risk assessment in LAM patients. Further studies with larger cohorts are warranted to validate these findings and explore their clinical applications in early diagnosis and personalized disease management.

## Figures and Tables

**Figure 1 ijms-26-06005-f001:**
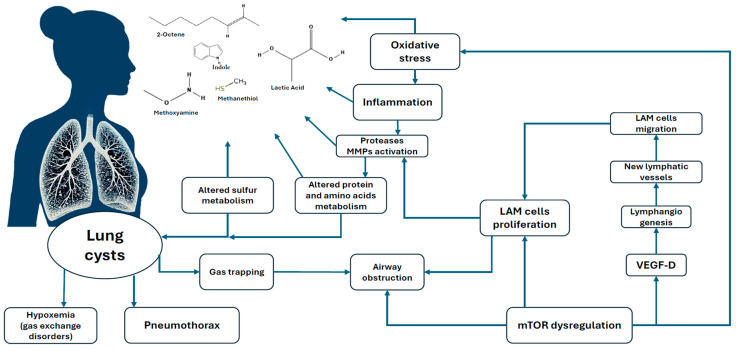
Proposed genesis of VOCs as a result of pathogenesis in LAM.

**Table 1 ijms-26-06005-t001:** Participants’ characteristics.

	LAM	Control	*p*-Value
Patients and controls (all women)	51	51	–
Age, years	48.7 ± 10.9	49.0 ± 17.9	0.536
BMI kg·m^2^	24.7 ± 5.1	25.9 ± 5.7	0.347
Smoking status			0.327
Never	50 (98.0)	49 (96.1)	
Former	0	2 (3.9)	
Current	1 (2.0)	0	
**mMRC, scores**	**1.5 ± 1.1**	**0.0 ± 0.1**	**<0.001**
**FVC % predicted**	**89.0 ± 18.7**	**100.5 ± 11.6**	**0.008**
**FEV_1_ % predicted**	**72.9 ± 28.6**	**101.6 ± 11.9**	**<0.001**
**FEV_1_/FVC %**	**63.8 ± 17.1**	**81.8 ± 6.3**	**<0.001**
**FEF_75_ % predicted**	**66.3 ± 44.2**	**141.0 ± 73.4**	**<0.001**
**RV % predicted**	**164.6 ± 53.8**	**118.7 ± 22.2**	**<0.001**
**FRC % predicted**	**131.6 ± 31.1**	**107.8 ± 20.6**	**<0.001**
**RV/TLC % predicted**	**143.1 ± 47.6**	**112.4 ± 21.1**	**<0.001**
**DLCO % predicted**	**62.8 ± 24.1**	**97.0 ± 13.5**	**<0.001**
**DLCO/VA % predicted**	**67.6 ± 22.6**	**101.6 ± 14.9**	**<0.001**
CT characteristics of the lungs			
Volume of lung cysts, % *	22.3 ± 16.5	NA	
B1 dist. average lumen area, mm^2^ **	1.4 ± 0.6	NA	
B10 dist. average lumen area, mm^2^	1.3 ± 0.6	NA	
History of extrapulmonary manifestations			
Pneumothorax	25 (49.0)	NA	
Pulmonary bleeding	13 (25.5)	NA	
Angiomyolipoma	31 (60.8)	NA	
Lymphangioleiomyoma	25 (49.0)	NA	
Chylothorax	13 (25.5)	NA	
Bronchodilators	27 (52.9)	NA	
Oxygen therapy	6 (11.8)	NA	
mTOR inhibitors	20 (39.2)	NA	

Data are presented as mean ± SD or number (%). Significant differences are highlighted in bold. * volume of cysts in both lungs; ** average lumen area of the distal part of the B1 and B10 bronchi of the right lung. BMI: body mass index, CT: computed tomography, DLCO: diffusing capacity of the lungs for carbon monoxide, FEV1: forced expiratory volume in 1 s, FEF75: the forced expiratory flow when 75% of FVC has been exhaled, FRC: functional residual capacity, FVC: forced vital capacity, mMRC: Modified Medical Research Council, mTOR: mammalian target of rapamycin, NA: not available, RV: residual volume, SD: standard deviation, TLC: total lung capacity, VA: alveolar volume.

**Table 2 ijms-26-06005-t002:** Predictors of LAM diagnosis by the XGBoost algorithm.

*m*/*z*	Name of the Chemical Substance	Mass Error, ppm	Feature Importances	
			Forced Expiratory Maneuver	Normal Quiet Breathing
44.99	NA		0.00677846	0.00683706
50.00	NA		0.00679701	0.00683653
82.07	NA		0.00681889	0.00683042
90.06	Lactic Acid	346.3 *	0.00679180	0.00683309
99.07	NA		0.00679757	0.00682748
113.13	NA		0.00685565	0.00691124
127.00	NA		0.00684542	0.00682354
150.10	NA		0.00683700	0.00685841
329.82	NA		0.00682365	0.00682728

* unresolved signal for *m*/*z* peak, presumably lactic acid. *m*/*z* —mass-to-charge ratio; XGBoost—gradient boosting classifiers

**Table 3 ijms-26-06005-t003:** Predictors of functional and clinical outcomes in LAM by the XGBoost algorithm *.

Clinical and Functional Endpoints	*m*/*z*	Name of the Chemical Substance	Mass Error, ppm	Feature Importances	
				Forced Expiratory Maneuver	Normal Quiet Breathing
Pneumothorax	103.08	Isopropyl acetate	−1.80	0.00695703	0.00701847
	133.10	2-Ethoxyethyl acetate	73.56	0.00756860	0.00716509
FEV_1_/FVC < LLN	63.02	Dimethyl sulfide	−110.04	0.00696004	0.00684350
	129.07	NA	–	0.00692103	0.00686908
	141.13	2-Nonenal	−13.31	0.00687696	0.00700937
FEV_1_ < LLN	48.04	O-methylhydroxylamine	−22.42	0.00692541	0.00684413
	173.15	Isomers of Oxononanoic acid	197.67	0.00679017	0.00688988
RV/TLC > LLN	74.05	N-methylacetamide	−138.07	0.00855004	0.0269816
RV > LLN	42.03	Acetonitrile	−124.36	0.00290517	0.00849818
DLCO < LLN	48.04	O-methylhydroxylamine	−22.42	0.00687678	0.00690884
Volume of lung cysts **	129.07	NA	–	0.00695566	0.00699561
B1 and B10 dist. average lumen area ***	57.06	Isobutene	−150.66	0.00722638	0.00729180
	73.06	Methyl Ethyl Ketone	−20.36	0.00720196	0.00705437
	91.06	2-Butanethiol	−34.75	0.00699667	0.00724412
	181.00	Oxidized Lipid Fragment	−198.72	0.00699667	0.00699890

* The table shows only annotated VOCs; more complete data are in Appendix A; ** volume of cysts in both lungs; *** the average lumen area of the distal part of the B1 and B10 bronchi of the right lung. DLCO: diffusing capacity of the lungs for carbon monoxide; FEV1: forced expiratory volume in 1 s; FVC: forced vital capacity; LLN: lower limit of normal (z-score < −1.645); RV: residual volume; TLC: total lung capacity; XGBoost: gradient boosting classifiers

**Table 4 ijms-26-06005-t004:** Direct comparison differences in the presence of VOCs in LAM vs. controls *.

Clinical and Functional Endpoints	*m*/*z*	VOCs Name **	Forced Expiratory Maneuver			Normal Quiet Breathing		
			LAM	Control		LAM	Control	
LAM diagnosis	44.99	NA	7558.6 ± 1437.1	8810.6 ± 2368.3	*p* < 0.001	7056.6 ± 1291.1	8035.1 ± 1770.8	*p* < 0.001
	50.00	NA	62.5 ± 8.5	70.5 ± 14.9	*p* < 0.05	64.0 ± 9.4	73.6 ± 22.3	*p* < 0.05
	82.07	NA	95.1 ± 29.3	98.9 ± 29.1	*p* < 0.05	52.8 ± 26.2	53.1 ± 27.8	*p* < 0.05
	90.06	Lactic Acid	58.4 ± 16.2	65.2 ± 20.0	*p* < 0.001	57.1 ± 17.1	60.0 ± 16.1	*p* < 0.05
	99.07	NA	76.7 ± 24.1	80.0 ± 25.9	*p* < 0.05	68.7 ± 19.8	72.0 ± 17.2	*p* < 0.05
	113.13	NA	51.6 ± 9.6	54.1 ± 9.0	*p* < 0.02	51.6 ± 9.0	56.7 ± 8.4	*p* < 0.03
	127.00	NA	59.4 ± 18.6	64.7 ± 24.5	*p* < 0.05	57.5 ± 18.7	60.1 ± 21.4	*p* < 0.001
	150.10	NA	28.8 ± 8.7	42.0 ± 85.2	*p* < 0.05	26.3 ± 6.4	33.5 ± 40.0	*p* < 0.05
	329.82	NA	1598.8 ± 521.9	1398.9 ± 562.5	*p* < 0.05	1626.3 ± 519.1	1411.8 ± 545.5	*p* < 0.01
Pneumothorax	103.08	Isopropyl acetate	109.0 ± 68.4	107.7 ± 69.5	*p* < 0.01	113.2 ± 72.5	103.2 ± 62.2	*p* < 0.001
	133.10	2-Ethoxyethyl acetate	32.4 ± 11.5	37.7 ± 9.9	*p* < 0.01	34.0 ± 7.2	35.6 ± 7.7	*p* < 0.1
	63.02	Dimethyl sulfide	26.1 ± 6.0	87.0 ± 61.6	*p* < 0.001	27.0 ± 11.4	83.6 ± 63.1	*p* < 0.001
FEV_1_/FVC < LLN	129.07	NA	133.6 ± 31.1	135.6 ± 87.0	*p* < 0.001	137.5 ± 102.4	139.7 ± 131.5	*p* < 0.05
	141.13	2-Nonenal	2401.3 ± 1690.9	2695.1 ± 1420.6	*p* < 0.05	2902.3 ± 1690.9	3145.4 ± 1829.0	*p* < 0.015
	48.04	O-methylhydroxylamine	138.1 ± 92.5	135.6 ± 87.0	*p* < 0.001	142.5 ± 102.4	139.7 ± 131.5	*p* < 0.01
FEV_1_ < LLN	173.15	Isomers of Oxononanoic acid	152.5 ± 115.4	94.8 ± 31.3	*p* < 0.001	109.6 ± 27.7	90.6 ± 28.8	*p* < 0.01
RV/TLC > LLN	74.05	N-methylacetamide	160.6 ± 21.8	153.9 ± 47.4	*p* < 0.001	148.6 ± 39.0	145.5 ± 42.4	*p* < 0.01
RV > LLN	42.03	Acetonitrile	76.1 ± 26.8	98.9 ± 29.1	*p* < 0.001	73.8 ± 27.7	91.0 ± 29.7	*p* < 0.05
DLCO < LLN	48.04	O-methylhydroxylamine	138.1 ± 92.5	135.6 ± 87.0	*p* < 0.05	147.5 ± 102.4	139.7 ± 131.5	*p* < 0.001
Volume of lung cysts ***	129.07	NA	51.7 ± 12.4	55.4 ± 10.4	*p* < 0.001	42.5 ± 8.1	50.9 ± 8.1	*p* < 0.001
B1 and B10 dist. average lumen area ****	57.06	Isobutene	624.7 ± 389.6	610.3 ± 378.8	*p* < 0.001	601.0 ± 355.5	547.0 ± 313.8	*p* < 0.001
	73.06	Methyl Ethyl Ketone	206.3 ± 66.1	215.2 ± 70.1	*p* < 0.001	196.5 ± 63.4	197.0 ± 53.6	*p* = 0.1
	91.06	2-Butanethiol	115.5 ± 35.0	127.1 ± 46.8	*p* < 0.001	108.8 ± 34.1	117.3 ± 41.8	*p* < 0.001
	181.00	Oxidized Lipid Fragment	31.9 ± 29.2	36.9 ± 25.7	*p* < 0.05	37.3 ± 40.8	39.7 ± 38.3	*p* < 0.05

* The Mann–Whitney U-test (for two groups) was used; data are presented as area comparisons (peak area for the target ion expressed as count per second); ** the putative chemical was identified using Ionicon libraries, the Human Metabolome Database and literature data; *** volume of cysts in both lungs; **** the average lumen area of the distal part of the B1 and B10 bronchi of the right lung. DLCO: diffusing capacity of the lungs for carbon monoxide; FEV1: forced expiratory volume in 1 s; FVC: forced vital capacity; LLN: lower limit of normal (z-score < −1.645); NA: not available; RV: residual volume; TLC: total lung capacity.

## Data Availability

The original contributions presented in this study are included in the article/Appendix A. Further inquiries can be directed to the corresponding author.

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
