# Peer review of "Exhaled Breath Analysis in Lymphangioleiomyomatosis by Real-Time Proton Mass Spectrometry"

_ijms, 2025, doi:10.3390/ijms26136005_

Round 1

Reviewer 1 Report

Comments and Suggestions for Authors

I would like to commend the authors for their innovative and timely contribution on the use of real-time proton-transfer-reaction mass spectrometry (PTR-TOF-MS) for the identification of volatile organic compounds in exhaled breath of patients with LAM. The study provides a promising perspective on non-invasive diagnosis and monitoring of this rare disease and demonstrates an impressive application of machine learning XGBoost for feature selection.

Despite the strengths, I would like to highlight some aspects that could be addressed or expanded in future work:

  • While 51 participants is considerable for a rare disease, the sample remains limited in terms of statistical validity. It would be valuable to know how the authors plan to extend the study to larger and more diverse cohorts to strengthen the reliability and applicability of their findings.
  • Given the novelty and potential clinical impact of this work, the conclusions appear too brief. I encourage the authors to elaborate on the challenges of implementation, and next steps for translation into practice, particularly considering the complexities of VOC analysis in breath.

  • The exclusive use of XGBoost is justified, but it would have been more robust to compare its performance with other classification algorithms to confirm the stability and accuracy of the predictive models. This is only a suggestion. 

  • A significant number of VOCs remained chemically unannotated. While this is understandable due to current database limitations, probably complementary methods such as GC-MS might help in identifying unknown but relevant metabolic markers and enhancing biological interpretation.

Author Response

Thank you very much for taking the time to review this manuscript. In the attached file you will find detailed responses and corresponding corrections in the resubmitted files.

Reviewer 2 Report

Comments and Suggestions for Authors

The manuscript submitted by Malika Mustafina et al. describes the exhaled breath analysis by PRT-MS-TOF for the identification of Lymphangioleiomyomatosis (LAM) biomarkers of disease that can potentially be used for LAM early detection and diagnosis. To my mind, the study in its presented form has some serious drawbacks and limitations, and I can’t recommend this manuscript for publication.

  1. The analysis of environmental VOCs content has not been carried out and their impact on breath VOCs pattern was not investigated and ambient air correction has not been done. The exhaled breath contains as known volatile organic compounds (VOC) that are distinguished as endogenous, related to the organism’s metabolic and biological processes, or exogenous, inhaled or ingested from the environment. Some indicated biomarkers are exogeneous, e.g., acetonitrile, azulene, or dual origin. Azulene is not a naturally occurring metabolite and is only found in those individuals exposed to this compound or its derivatives (HMDB Human Metabolome Database: Showing metabocard for Azulene (HMDB0248822). Thus, variation in the composition of the environmental VOCs (e.g. changing the room of sampling) can affect the results of analysis and, finally, the diagnosis.
  2. The authors claim that the suggested approach demonstrated “high diagnostic accuracy”, but they succeeded only in discrimination of LAM patient from healthy controls, but not from patients with other lung diseases. This claim demonstrates the overinterpretation of research findings.
  3. More details on instrumental conditions and VOC identification should be presented so that to demonstrate if identification is valid. Drif voltage? drift pressure? E/N value? What MS databased were used for VOC identification and what was the matching score? Other hits if any? The authors mention “Ionicon ion libraries - one containing 300 factory-calibrated masses and 325 the other comprising 1000 factory masses”, but these are very few compounds for the detected features identification. Without clear information for the features identification details, VOC identification can be doubtful, e.g. identification of methinophosphide - an unstable and highly reactive phosphorus species. Methinophosphide is not listed in the Human Metabolome Database or KEGG PATHWAY Database, and the assumption that it is a new human metabolite must be thoroughly substantiated.
  4. Another new biomarker indicated by the authors is tetramethylstannane with m/z=181. As known, tin has ten stable isotopes with the most abandoned 120 (32,6%), 118 (24,2%), 116 (14,5%), 119 (8,59%). Have the corresponding isotopomers been detected? If yes, why are they not mentioned and listed in the Table? It could confirm that tetramethylstannane was indeed detected in the breath. According to the data presented in table 3 the m/z uncertainty for this feature is 0,0048m/z and other hits can correspond.

Author Response

(The authors gave the same response as above.)

Reviewer 3 Report

Comments and Suggestions for Authors
  1. The manuscript highlights several VOCs as indicators of LAM, yet the biological relevance of compounds such as methinophosphide and azulene in LAM pathophysiology is still uncertain. Offering insights into potential metabolic origins or pathways could enhance the clinical understanding.
  2. In Tables 2 and 3, numerous m/z features are marked as "NA." To improve the manuscript's rigor, it would be beneficial to explain the criteria or databases used for identifying VOCs and clarify whether tentative matches were included or excluded.
  3. The introduction connects LAM pathogenesis with mTOR signaling, but the possible link between mTOR-induced cellular alterations and specific VOC emissions remains largely unexplored. Including a brief mechanistic discussion or hypothesis here could enhance the depth of the content.
  4. Although breath samples were obtained through both quiet and forced maneuvers, the study lacks information on aspects like the duration of sampling, instructions for breath-holding, or measures for ensuring reproducibility. Including these details would enhance the methodological reproducibility of the study.
  5. Some of the key VOCs identified, such as methanethiol and lactic acid, have been previously associated with lung diseases in earlier research. Including a brief discussion in the manuscript that compares these results with existing literature on breath biomarkers could enhance the work by emphasizing its novelty or its role in confirming previous findings.
  6. While Tables 2 and 3 offer valuable information, they are quite complex. To enhance data visualization and engage readers more effectively, incorporating a heatmap or bar chart that summarizes the top 10 VOCs across clinical endpoints, such as pneumothorax and FEV1/FVC, would be beneficial.
  7. Although several VOCs are highlighted as key predictors, the criteria for their selection, such as statistical thresholds and effect sizes, are not explicitly detailed. Providing a clearer explanation of the methodology used to deem these predictors as "significant" would enhance the study's rigor and reproducibility.
  8. The limitations section is organized well, yet it could be improved by offering a more explicit discussion on how these constraints may have impacted certain outcomes or interpretations, such as the effects of treatment on VOC composition. Furthermore, proposing ways for future research to overcome these limitations would enhance the section's depth.

Author Response

(The authors gave the same response as above.)

Reviewer 4 Report

Comments and Suggestions for Authors

In this manuscript(ijms-3559025-peer-review-v1), real-time proton mass spectrometry was used to successfully identify specific biomarkers of LAM in exhaled breath. These findings provide valuable insights into LAM-related metabolic changes and highlight the potential of non-invasive respiratory analysis as a diagnostic tool. In addition, predictors of complications were identified, including pneumothorax, impaired respiratory function, and characteristic radiological findings, such as lung cysts and a decrease in the average lumen area of the distal bronchus shown on computed tomography. These biomarkers may serve as valuable indicators for disease monitoring and risk assessment in LAM patients. But published in the International Journal of Molecular Sciences, it still needs to be revised.

Some detail information need to be clarified:

1.In the identification of volatile organic compounds as LAM predictors in Article 2.2, the XGBoost algorithm was used to determine the predictors of LAM.Whether the algorithm should be introduced in the previous article and the basis for determination.

2.Methoxyphosphine, methanthiol, methyl hydroxylamine, azulene. The reasons for these most important volatile organic compounds identified as predictors are not well explained in this paper.

3.The exhaled breath compared in this paper is for LAM.Whether these gases appear in other diseases and whether they are specific

Author Response

(The authors gave the same response as above.)

Round 2

Reviewer 2 Report

Comments and Suggestions for Authors

I believe that the major flaws in this study are not addressed after revision because of an initial problem with the study design, namely:
I. Correcting for ambient air in PTR-MS breath analysis is crucial to ensure accurate measurements of volatile organic compounds (VOCs) [1,2] and this has not been done. 
1.Roquencourt C, Grassin-Delyle S, Thévenot EA. ptairMS: real-time processing and analysis of PTR-TOF-MS data for biomarker discovery in exhaled breath. Bioinformatics. 2022 Mar 28;38(7):1930-1937.
2. Camille Roquencourt et al 2024 J. Breath Res. 18 016006
II. There is no new evidence that the suggested method can discriminate the LAM patients from patients with other lung diseases. The mentioned in authors' response  review I believe could hardly solve the question because patients samples were analysed at different points in time.

Author Response

Thank you very much for taking the time to review this manuscript. Please see the attachment.

Round 3

Reviewer 2 Report

Comments and Suggestions for Authors

The authors' answers are not convincing, and I can’t recommend this manuscript for publication. The study has severe drawbacks in research design.

  1. The correction for ambient air has not been done and this is crucial for the detection of endogenous metabolites specific to pathophysiology of the disease and their differentiation from inhaled environmental exogenous compound (sampling room air that is variable). The background signal referred by the authors has nothing to do with ambient (room) air spectrum.  
  2. I repeat once again that there is no evidence that the suggested method can discriminate the LAM patients from patients with other lung diseases and thus this method at the presented stage of development has low value for the future clinical application. The authors should demonstrate that PTR-TOF-MS breath analysis has the distinctive power to discriminate LAM patients from health people and patients with other lung diseases.

Additional remarks:

  1. The authors identify feature with m/z 129.07 as indole (Table 3). But indole has exact mass 117.06 and protonated adduct 118,065. How can feature with m/z 129 be identified as indole?
  2. Feature with m/z 113.13 is identified as 2-octene. Why not 1-octene?
  3. Can authors comment if the presented predictors are endogenous metabolites?
  4. The authors don’t indicate if the claimed predictors are increased or decreased in the breath of LAM patients in comparison to healthy controls.
